# Cholera in Kenya: A scoping review of current research, evidence gaps and future directions

Kevin Wamae [1]*, John Magudha [1], Agnetor Kakungu [1], Steve Aricha[2], Daniel Langat[3], Samson Kinyanjui [1,4], Jolynne Mokaya [5], Nicholas R. Thomson [5], Charles Agoti [1], George Githinji [1]

1 KEMRI-Wellcome Trust Research Programme, Kilifi, Kenya, 2 National Genomics and Molecular Surveillance Laboratory, Ministry of Health, Nairobi, Kenya, 3 Division of Disease Surveillance and Response, Ministry of Health, Nairobi, Kenya, 4 Centre for Tropical Medicine and Global Health, Nuffield Dept of Medicine, University of Oxford, Oxford, United Kingdom, 5 Wellcome Sanger Institute, Wellcome Genome Campus, Hinxton, Cambridge, United Kingdom

* wamaekevin@gmail.com

## Abstract

Cholera remains a major public health challenge in Kenya, driven by environmental pollution, poor water, sanitation, and hygiene (WASH) facilities, weak monitoring systems, and climate effects. While there have been improvements in studying the disease, testing, monitoring, and environmental checks, the spread continues, especially in informal settlements, rural areas, and refugee camps. This review looked at 106 peer-reviewed studies published between 1979 and 2024. It aimed to describe the scope, progress, and gaps in cholera research in Kenya. Five databases, including Google Scholar, Web of Science, PubMed, Embase, and Scopus, were searched using the terms "cholera" and "Kenya." Titles and abstracts were reviewed using Rayyan, and data were collected using a standard form that noted study goals, methods, findings, limitations, and geographic coverage. The evidence was analysed thematically, and trends were tracked over time. Among 845 records found, 106 met the criteria for inclusion. The literature consistently connects cholera outbreaks in Kenya to climate events like flooding, drought, and El Niño, with greater vulnerability seen in informal settlements and refugee areas. Surveillance systems are still disconnected, often leading to delays in reporting and limited information at sub-county levels. Assessments of the health system reveal ongoing issues, including a shortage of laboratory resources and necessary tests. Despite innovations such as rapid diagnostic tests, whole-genome sequencing, and spatial modelling that have enhanced outbreak detection and understanding of transmission, their regular use is hindered by cost, infrastructure, and workforce issues. Social and behavioural factors, such as low-risk perception and gaps between knowledge and practice, also contribute to continued transmission. In summary, controlling cholera in Kenya needs a coordinated, multi-faceted approach. This should focus on strengthening surveillance, improving WASH facilities, increasing laboratory and molecular capabilities, expanding vaccine

**Data availability statement:** All data and related metadata underlying these findings are reported in the paper and its Supporting Information files.

**Funding:** This work was supported by a Sub-Saharan Africa Consortium for Advanced Biostatistics (SSACAB) fellowship, awarded to Kevin Wamae [D2304120-01], and funded by the Science for Africa Foundation with support from Wellcome Trust and the UK Foreign, Commonwealth & Development Office. The funders had no role in study design, data collection and analysis, decision to publish, or preparation of the manuscript.

**Competing interests:** The authors have declared that no competing interests exist.

access, and fostering community-led efforts to turn scientific progress into effective public health measures.

## 1. Introduction

Cholera is an acute diarrhoeal disease caused by the Gram-negative bacterium *Vibrio cholerae* [1] and remains a significant public health problem in Kenya. Between January 2022 and December 2024, over 12,666 cases and 209 deaths were reported nationwide [2]. Since the first documented outbreak in 1971, cholera has disproportionately affected populations in informal settlements, refugee camps, and rural areas, placing sustained strain on health systems and community resources [3,4].

Kenya's cholera burden is closely linked to unsafe drinking water, inadequate sanitation, climate variability, and socioeconomic inequalities that amplify transmission and limit access to timely care [5,6]. These challenges align with regional and global priorities, including the Global Task Force on Cholera Control's 2030 Roadmap, which targets a 90% reduction in cholera deaths, and commitments by African Heads of State to strengthen cross-border coordination and expand access to oral cholera vaccines [7,8].

Over the past four decades, cholera control efforts in Kenya have evolved from a primary focus on descriptive epidemiology and clinical management to more integrated approaches incorporating molecular tools, environmental surveillance, and community-based interventions [9,10,11,12]. However, persistent gaps in laboratory capacity, surveillance coordination, health-system preparedness, and equitable service delivery continue to undermine effective outbreak prevention and response [13,14].

This scoping review synthesises evidence from 106 studies published between 1979 and 2024 to map the landscape of cholera research in Kenya, assess progress made, identify persistent gaps, and inform future research and policy priorities.

## 2. Methods

### 2.1 Search strategy

The review follows the guidelines set by the Preferred Reporting Items for Systematic Reviews and Meta-Analyses extension for Scoping Reviews (PRISMA-ScR, S1 PRISMA Checklist) [15]. To identify relevant studies and ensure wider coverage across disciplines and publication types, we systematically searched five scientific databases: Google Scholar, Web of Science, PubMed, Embase, and Scopus. The search strategy used the terms "*cholera AND Kenya*" to capture all studies focusing on cholera epidemiology, surveillance, prevention, and control in the Kenyan context. The final database searches were conducted on 8th October 2024.

### 2.2 Inclusion and exclusion criteria

The identified articles were uploaded to Rayyan, a web platform for systematic reviews [16]. Studies were included if they addressed cholera-related research, were peer-reviewed, published in English, and focused specifically on cholera in Kenya. Studies focusing on pathogens other than *V. cholerae* or populations outside Kenya were excluded.

### 2.3 Screening and data extraction

Titles and abstracts were independently screened for relevance by two authors (KW and JM). Any conflicts in the decision to include or exclude a study were resolved through discussion and consensus with a third author (AK). Data from all included studies were extracted into a standardised form by KW. This extracted data was then independently verified for accuracy and completeness by JM and AK. The standardised form captured details such as title, problem statement, research gaps, objectives, methods used, results, conclusions, challenges, limitations, and the relevant themes. The results were subsequently synthesised by the full research team to provide an overview of existing cholera research in Kenya and highlight areas requiring further investigation.

### 2.4 Protocol registration and quality assessment

As this is a scoping review, the protocol was not formally registered with PROSPERO [17], as this is typically reserved for systematic reviews and is not a mandatory requirement for scoping reviews. Furthermore, we did not conduct a critical appraisal or quality assessment of the included studies, as the primary objective of a scoping review is to map the breadth of the literature rather than to synthesise findings or evaluate the risk of bias of individual studies [18].

## 3. Results

### 3.1 Characteristics of included studies

Initially, 845 records were retrieved from the databases. After removing 452 duplicates, 393 records remained for screening. Screening of titles and abstracts identified 230 studies eligible for full-text review, of which 192 were assessed for eligibility. In total, 106 studies published between 1979 and 2024 were included in the review (**Fig 1**). Key characteristics of all included studies, such as publication details, study objectives, methodological approaches, geographic settings, thematic classifications, principal findings, identified research gaps, and reported limitations, were systematically charted and are presented in S1 Table.

The included studies spanned a wide range of research domains. Epidemiology was the most prevalent theme (31 studies), followed by environmental factors (23) and antimicrobial resistance (21). Substantial attention was also given to water, sanitation and hygiene (WASH) and genomics (17 studies each), health systems (16), and surveillance (12). Additional themes included diagnostics and social science (7 studies each), clinical management (6), health education and natural products (4 each), immunology and disease modelling (2 each), and smaller contributions to drug development and health economics (1 study each). Many studies addressed multiple themes, reflecting the complex and interconnected nature of cholera research in Kenya.

### 3.2 Geographic distribution and methodological approaches

Although all research incorporated data from Kenya, the geographic scope exhibited significant variation. The majority (n = 79) concentrated solely on Kenya, while roughly one-quarter (n = 27) incorporated data from other nations, especially in East Africa. The research employed several methodological approaches, including laboratory studies (n = 36), epidemiological investigations (n = 59), environmental sampling (n = 18), social science research (n = 5), genetic analysis (n = 16), and health systems evaluation (n = 10). The study designs were mainly quantitative (n = 77), mixed-methods (n = 11), reviews (10), and qualitative studies (8).

### 3.3 Temporal trends

From 1979 through the 1990s, cholera research in Kenya mainly focused on antimicrobial resistance, genomics, and epidemiology. In the early 2000s, research topics broadened. Environmental factors and water, sanitation, and hygiene (WASH) began to appear consistently after 1999 and 2001, respectively. Meanwhile, interest in surveillance and health

Global Public Health
PLOS

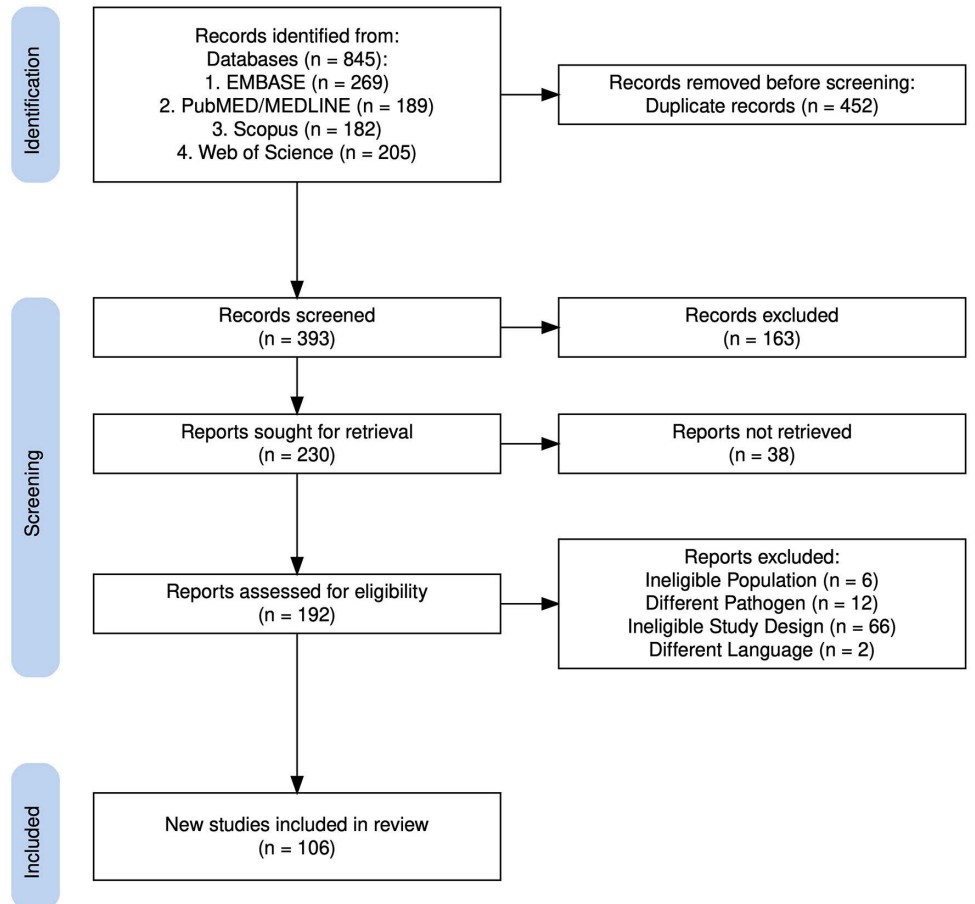

**Fig 1. PRISMA flow diagram of study selection for the cholera scoping review in Kenya.** A total of 845 records were identified through searches of five electronic databases. After removal of 452 duplicate records, 393 titles and abstracts were screened, of which 163 were excluded. Full-text reports were sought for 230 records, and 38 could not be retrieved. Of the 192 reports assessed for eligibility, 86 were excluded due to an ineligible study design (e.g., editorials, commentaries, protocols, or other non-research articles; n = 66), focus on non-Kenyan populations (n = 6), investigation of pathogens other than Vibrio cholerae (n = 12), or publication in a non-English language (n = 2). In total, 106 studies met the inclusion criteria and were included in the final analysis. The figure was generated using the PRISMA 2020 R package (v1.1.3) [19].

systems increased, with both themes receiving more attention from the early 2000s onward. After 2010, research topics began to diversify noticeably as molecular and modelling approaches gained importance. Genomics and antimicrobial resistance also experienced a significant revival, and studies in diagnostics, disease modelling, and immunology grew larger. From 2020 to 2024, cholera research shifted toward a more integrated and interdisciplinary approach. Studies now increasingly include social science, health education, and health economics, alongside a strong and consistent focus on environmental factors, WASH, and epidemiology (**Fig 2**).

### 3.4 Thematic areas

Having categorised the identified studies into key themes, overlapping thematic areas were consolidated to support synthesis. This section highlights major findings, research contributions, and persistent gaps across the consolidated themes.

   **3.4.1 Epidemiology, surveillance, and disease modelling.** Research in this thematic area focuses on describing disease incidence and risk patterns, evaluating the performance and limitations of surveillance systems, and applying

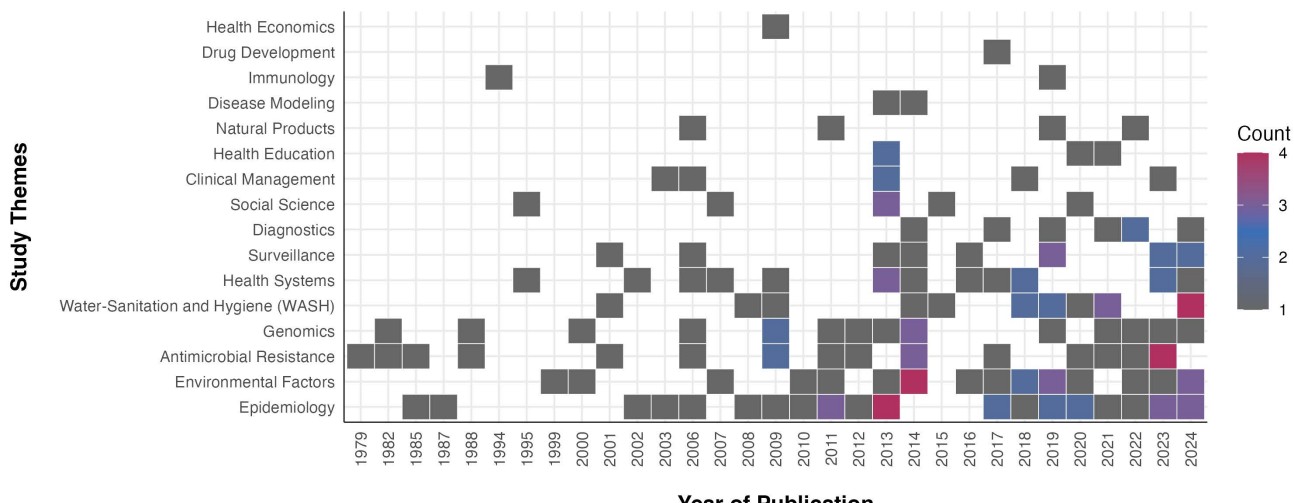

**Fig 2. Temporal Trends in Peer-Reviewed Cholera Research in Kenya by Theme, 1979-2024.** This heatmap displays the annual frequency of peer-reviewed cholera studies conducted in Kenya between 1979 and 2024, organised across sixteen research themes. Each square represents at least one publication in each year-theme combination, with colours indicating study counts: light grey denotes a single publication, blue indicates moderate research activity (two to three studies), and maroon represents the highest annual counts (four or more studies). The figure demonstrates the evolution in research emphasis, with early work dominated by epidemiology, antimicrobial resistance, and genomics. From the early 2000s, research activity expanded into water, sanitation and hygiene (WASH), environmental factors, surveillance, and health systems. The most recent decade shows further diversification into immunology and disease modelling, alongside increasing attention to socio-economic domains such as health economics and social science.

statistical and spatial modelling approaches to characterise transmission dynamics, identify hotspots, and inform targeted interventions

**Surveillance and data patterns:** Research conducted between 1997 and 2022 examined the epidemiology and risk factors of cholera in Kenya using various methodological frameworks. National surveillance data from 1997-2010 recorded 68,522 suspected cases and 2,641 deaths, with peak incidence during the 1997–1999 and 2007–2009 periods [20]. These outbreaks primarily affected children under 15 years of age and populations in designated high-risk areas. A 2009 case-control study in the Kakuma refugee camp identified handwashing with soap as a protective factor, while the use of dirty water containers was identified as a risk factor [21]. Statistical modelling of trends from 2007 to 2022 indicated an increase in both the frequency and geographic distribution of recorded outbreaks [22].

**Data limitations and identification methods:** Studies identified several factors contributing to the underestimation of disease burden, including underreporting of deaths occurring outside health facilities and inconsistent demographic data collection. Historical case definitions excluded children under five until an outbreak was bacteriologically confirmed [21,20,22]. In the Dadaab and Kakuma refugee camps, case verification was further limited by insecurity and personnel shortages [23,24]. There have been technical developments in data collection including the use of multiplex molecular detection methods [25] and hotspot mapping based on mean annual incidence, persistence, and WASH indicators [3].

**Risk patterns and hotspots:** Between 2015 and 2020, Kenya reported over 30,000 suspected cholera cases across 32 counties. Hotspot analysis using the Global Task Force on Cholera Control (GTFCC) methodology indicated that 3.2 million people could be misclassified when using only county-level data rather than sub-county data [26]. Persistent outbreaks were recorded in refugee camps, characterised by high population densities and continuous influxes [27,21,23,24,28]. Conversely, rural and peri-urban outbreaks were linked to seasonal flooding and contaminated water sources, with evidence suggesting that different *V. cholerae* strains may drive distinct epidemic patterns across these settings [10].

**Environmental and behavioural hazards:** Specific hazards associated with outbreak risk include the sharing of unchlorinated water containers [29], consumption of surface water [30], (Onyango et al. 2013), and participation in communal meals at major events [31,32]. In informal urban settlements, risk factors included poor sanitation and the use of uncontrolled water vendors [33,34]. In refugee settings, recent arrival at the camp was positively associated with an increased risk of infection [35,24].

**Modelling approaches:** Studies utilised Agent-Based Models (ABM) to simulate dynamic interactions between individuals and the environment using a Susceptible-Exposed-Infected-Recovered (SEIR) framework [36,37]. When integrated with Geographic Information Systems (GIS), these models captured individual-level behaviours and spatial dynamics of disease spread via shared water sources. Additionally, Poisson space-time scanning was applied to identify and target specific geographic clusters and hotspot locations [38].

### 3.4.2 Environmental determinants and Water, Sanitation, and Hygiene (WASH).

Environmental conditions and deficiencies in Water, Sanitation, and Hygiene (WASH) infrastructure significantly influence cholera transmission in Kenya. Studies in this thematic area examine contaminated water sources, environmental reservoirs, and climate variability, alongside structural and behavioural WASH gaps, to explain the persistence of *V. cholerae* and the amplification of outbreaks across diverse ecological and settlement contexts.

**Contaminated water sources:** Cholera outbreaks are frequently associated with water sources contaminated by faecal matter in urban informal settlements, flood zones, sewers, and irrigation canals [12]. While groundwater and lake water used for consumption or fish-processing often fail to meet safety standards, piped water systems are also subject to contamination via illegal connections and infrastructure deterioration [39]. Research has documented the presence of the pathogen in various water points, emphasising a correlation between water quality and disease incidence.

**Environmental reservoirs:** Large water bodies, such as Lake Victoria, serve as reservoirs for *V. cholerae*, with water hyacinth blooms specifically linked to increased epidemic frequency [40]. Isolates recovered from drainage systems, flood zones, and irrigation canals suggest the existence of multiple environmental niches that sustain the pathogen between active outbreaks [41,42,43]. However, a 2024 multi-country genomic analysis of 728 *V. cholerae* O1 isolates indicates that cross-border human mobility is a more significant driver of spread in Africa than long-term environmental persistence or localised pathogen evolution [44].

**Ongoing environmental investigations:** Current research examines the role of bacteriophages lytic to *V. cholerae* in Kenyan waterways as potential natural regulators of pathogenicity [45]. Additionally, studies have focused on the contribution of surface water run-off, zooplankton, and various fish species in the dissemination of the disease [46,47].

**Climate variability:** Extreme weather events, including both floods and droughts, correlate with increased cholera risk. While fluctuations in water levels promote plankton blooms that support bacterial growth, heavy precipitation events result in the wash-off of pathogens into water systems and the overflow of sewage infrastructure [48,49]; (Anyamba et al. 2019). Modelling of these climatic trends has been used to develop frameworks for early-warning systems and environmental monitoring [36,50].

**WASH infrastructure gaps:** Persistent cholera transmission is documented in regions characterised by inadequate sanitation facilities, high rates of open defecation, and limited access to water treatment [39,51,52]. Research shows that progress in hygiene is often offset by the use of unsealed water containers, leaking sewage lines, and poorly maintained latrines [53,12]. Conversely, studies have recorded significant risk reduction associated with handwashing practices and the use of point-of-use chlorination [54,53].

### 3.4.3 Health systems capacity and outbreak response.

Effective cholera outbreak response in Kenya is shaped by the performance of surveillance systems, the preparedness of health services, and the strength of coordination mechanisms across sectors and administrative levels. The literature in this area documents both improvements in reporting and case management capacity and persistent systemic challenges, including data quality gaps, resource constraints, and coordination failures that undermine timely and effective response.

**Surveillance systems and data quality:** The implementation of the Integrated Disease Surveillance and Response (IDSR) framework in Kenya has been associated with improved reporting, however, underreporting and data discrepancies persist [26,22]. During the 2008 post-election period, active case finding recorded a 200% higher mortality rate compared to passive surveillance techniques [55]. Factors identified as limiting real-time monitoring include inconsistent application of case definitions, personnel shortages, network connectivity issues, and notification delays [24,14]. Additionally, incomplete sub-county data has been shown to result in the misclassification of risk levels [26]. Variations in reporting quality are also noted between different counties [22].

**Health system preparedness:** Research indicates that service deficiencies persist despite increased access to oral rehydration solutions (ORS) [13]. Delayed care-seeking among disadvantaged populations is linked to financial barriers and low health literacy [56]. In rural and refugee settings, such as Dadaab, studies have recorded shortages in staffing, medical supplies, and case management capacity [23].

**Coordination and response:** Outbreak response efforts are reported to be affected by challenges in leadership and the distribution of sectoral responsibilities [13]. The National Multi-Sectoral Cholera Elimination Plan (NMCEP 2022–2030) establishes multisectoral coordination structures at national and county levels, including County Cholera Elimination Steering Committees and a Risk Communication & Community Engagement (RCCE) pillar [6]. Historical data show that resource misallocation and civil unrest have contributed to delays in supply delivery and increased mortality rates [55,57]. Conversely, studies on hotspot mapping and immunisation campaigns report that multi-agency cooperation is a factor in successful intervention outcomes [3].

### 3.4.4 Diagnostics and clinical case management.

Research on diagnostics and clinical case management in Kenya emphasises the role of timely and accurate diagnosis, laboratory capacity, and effective case management in reducing cholera-related morbidity and mortality. Studies in this area assess the performance of rapid diagnostic tests and molecular methods, document gaps in laboratory and antimicrobial-resistance testing capacity, and examine the availability of essential treatment resources and infection-prevention practices in clinical settings.

**Diagnostic advances:** Rapid diagnostic tests (RDTs), such as Crystal VC-O1, have demonstrated 97.5% sensitivity and 100% specificity in the Kenyan context [58]. In resource-constrained settings, studies have utilised PCR confirmation using DNA recovered from RDT dipsticks as a cost-effective method for resolving diagnostic discrepancies [59]. For pediatric investigations, serological techniques and TaqMan Array Cards are reported to assist in early detection and outbreak tracking [60,25].

**Laboratory capacity and antimicrobial resistance:** Reports indicate that many healthcare institutions lack molecular platforms and antimicrobial-resistance (AMR) testing capabilities beyond simple dipstick assays [61,62,13]. These capacity gaps are most pronounced in refugee and rural health facilities [23,14]. High levels of antibiotic resistance have been documented, with one trial identifying that 77.3% of children received antibiotics without clinical necessity [30,63,64]. Alternative surveillance methodologies currently under investigation include filter-paper preservation for sample transport [65] and wastewater-based metagenomics [66].

**Clinical management and preparedness:** Early therapy and timely hospital admission are associated with improved clinical outcomes [55]. However, resource shortages have been recorded, for example, during the 2014–2016 period, 82% of hospitals maintained ORS stocks and 65% had available IV fluids [13]. Limited diagnostic availability is also linked to a higher risk of unrecognised co-infections [67]. Care-seeking behaviour is reported to be delayed by financial barriers and low community awareness of cholera symptoms [67].

**Infection control:** Studies have identified hygiene lapses within clinical settings, such as low levels of mobile phone sanitation among healthcare professionals, as factors that undermine infection prevention [68]. The literature suggests that growing antimicrobial resistance increases the reliance on rigorous cleanliness protocols and the identification of co-infections at the point of care [63,67].

### 3.4.5 Molecular epidemiology, vaccines and antimicrobial resistance.

Advances in whole-genome sequencing, molecular surveillance, and serological approaches have reshaped understanding of *V. cholerae* transmission, lineage evolution, and antimicrobial resistance in Kenya. The literature in this thematic area documents the circulation of pandemic and locally evolved lineages, the role of mobile genetic elements in resistance dissemination, and the implications of these findings for vaccine deployment and future therapeutic strategies.

**Molecular and Sero-epidemiology:** Genomic technologies have documented the evolution and dissemination of *V. cholerae* lineages within the region. Early research identified clonal expansions and the dominance of the El Tor biotype [69,70,71]. Genomic surveillance has further revealed the presence of "hybrid" El Tor variants that combine traditional virulence determinants with environmental resilience [72,73].

The ongoing seventh pandemic El Tor lineage (7PET), which originated in Indonesia in 1961 and subsequently spread from the Bay of Bengal into Africa in multiple waves, remains the globally dominant lineage [9]. High-resolution genomics confirmed that this lineage was responsible for recent large-scale African epidemics, including the 2022–2023 outbreak in Malawi [74]. WGS has also identified co-circulating *V. cholerae* lineages and the role of mobile genetic elements, such as integrative conjugative elements (ICE) and plasmids, in the dissemination of antimicrobial resistance [75,76,77,65]. Additionally, genomic analysis of isolates from Lake Victoria and coastal Kenya identified unique, locally evolved lineages that possess lower frequencies of virulence genes but are occasionally detected in human populations [41,42]. Although relatively few studies have employed serology to study cholera, antibody profiling has been identified as a promising tool for monitoring population-level exposure [60].

**Antimicrobial resistance:** *V. cholerae* strains in Kenya have demonstrated increasing resistance to several antibiotics, including tetracycline, streptomycin, and trimethoprim-sulfamethoxazole [69,78,75,79,77]. Studies report that this resistance is frequently propagated via plasmids and other mobile genetic elements [76,72].

**Vaccine factors and drug discovery:** While oral cholera vaccines (OCVs) are reported as effective, the literature identifies several barriers to use, including procurement costs, limited awareness, local perceptions, and global supply shortages [80,81,82]. Research into alternative therapeutic approaches has explored phage therapy and soil-derived antimicrobials, which have shown activity against resistant *V. cholerae* strains in laboratory settings [83]; (Maina et al. 2021).

### 3.4.6 Intervention studies and control strategies.

Evidence from Kenya evaluates the effectiveness of a range of cholera control interventions, including WASH improvements, vaccination campaigns, biological control approaches, and community-based behaviour change strategies. Studies in this area assess both the short-term and sustained impacts of these interventions across rural, urban, and refugee settings, highlighting contextual factors that influence implementation success and intervention uptake.

**Water treatment and sanitation:** In Maasai villages, solar disinfection was recorded to reduce cholera risk among children, with three cases reported in the treatment group (n = 155) compared to 20 cases in the control group [84]. However, other studies have reported inconsistent impacts of household water treatment efforts, particularly in refugee camps and informal settlements where transmission has been documented to persist despite infrastructure upgrades [85,86,28].

**Biological control strategies:** Research has identified bacteriophages isolated from Lake Victoria with lytic activity against both clinical and environmental *V. cholerae* strains [45], (2021). These studies characterise phages as a potential antimicrobial alternative for reducing pathogen concentrations in environmental reservoirs.

**Community education and behaviour change:** Data from Nyanza Province indicates a discrepancy between cholera awareness and household practice; despite high knowledge of symptoms, fewer than 20% of surveyed households had detectable chlorine levels in their stored water [87,80]. Educational interventions have been associated with short-term improvements in hygiene practices, but longitudinal evidence shows limited sustained impact [86].

**Vaccine acceptance and implementation:** In Western Kenya, survey data showed a 93% hypothetical acceptance rate for free oral cholera vaccines (OCV). However, willingness to participate decreased as out-of-pocket costs increased,

identifying affordability as a primary barrier to uptake [80,81]. The literature suggests that the integration of WASH services with OCV campaigns is associated with higher intervention uptake [85].

**Refugee camp interventions:** In the Kakuma and Kalobeyei settlements, incidence rates remain highest among children under five. Studies in these settings indicate that the effectiveness of integrated WASH and OCV interventions is contingent upon consistent funding and maintenance of infrastructure [85,28].

**3.4.7 Social, behavioural, and economic aspects of cholera.** Research on the social, behavioural, and economic dimensions of cholera in Kenya examines how knowledge, perceptions, cultural beliefs, and economic constraints shape prevention practices, care-seeking behaviour, and disease burden. The literature documents persistent gaps between awareness and practice, urban-rural disparities, and the financial impacts of cholera on households and the health system.

**Knowledge and practice gaps:** Research shows a disparity between high levels of cholera awareness and the implementation of preventive actions. In Isiolo, 99.3% of survey respondents were aware of cholera, yet only 50% reported treating their water, and fewer than 50% practised recommended sanitation behaviours [11]. Similarly, in Nyanza, while 80% of participants could identify symptoms, fewer than 20% maintained adequate chlorine levels in stored water [87]. An analysis of Kenyan school science textbooks found that while water safety and disease prevention are covered, there is a lack of detailed content on hygiene and sanitation [88]. Historical data from rural areas also suggests that while older adults demonstrate high symptom recognition, younger demographics are less likely to act on this knowledge [89].

**Urban-rural disparities:** The perception of cholera drivers varies by setting. In rural areas, the disease is primarily viewed through an environmental lens, whereas in urban settings, financial strain is reported as a primary concern [89]. These differing perspectives correlate with health behaviours; for example, health survey refusal rates were found to be significantly higher in urban areas (18.7%) compared to rural areas (6.8%) [90]. Studies also document the persistence of supernatural explanations for illness and the role of rapid urbanisation in creating informal settlements with limited service access [91,92].

**Economic impact:** Cholera is reported to impose a significant financial burden on the national health system. Estimates for the WHO African Region place the cost per case between $311 and $513, primarily attributed to lost productivity and premature mortality [93]. Households in rural areas are identified as particularly vulnerable to these economic shocks due to limited financial buffers [90].

**Sociocultural barriers:** Factors such as limited latrine availability, inadequate water treatment, and specific community norms are documented as impediments to prevention. In Lake Victoria basin communities, poverty-driven reliance on unsafe water sources is a persistent risk factor [94]. Furthermore, the literature identifies stigma and cultural taboos as factors that delay formal care-seeking [11], while school-based health messaging is sometimes reported to be misaligned with local cultural beliefs [88].

**3.4.8 Alternative and natural approaches to cholera control.** Studies in Kenya have investigated alternative and natural approaches to cholera control, including plant-derived compounds, traditional medicinal practices, and biological agents such as bacteriophages and soil-derived microorganisms. This body of literature primarily evaluates in vitro antimicrobial activity and explores the potential role of these approaches as locally accessible or complementary strategies, while also highlighting significant gaps in clinical validation and safety assessment.

**Natural product evaluation:** Several indigenous botanical extracts have demonstrated antibacterial activity against *V. cholerae* in laboratory settings. Kenyan tea extracts were reported to have larger inhibition zones compared to Nigerian varieties, a result attributed to higher phytochemical concentrations [95]. *In vitro* studies of *Moringa oleifera* and *Moringa stenopetala* methanol extracts showed a significant reduction in bacterial growth, particularly at a 40% concentration [96]. Furthermore, a review of 105 Kenyan medicinal plants identified multiple species with antimicrobial potential against waterborne pathogens [97].

**Traditional medicine use:** In rural Kenyan settings, traditional medicine is frequently utilised as a primary source of care. While these herbal treatments are noted for their accessibility and affordability, the literature indicates a lack of systematic clinical evaluation for their efficacy and safety [97]. Most existing research focuses on single-plant extracts, although community practice often involves the use of multi-herb remedies [96].

**Alternative control strategies:** Soil-derived *Actinomycetes* have been identified as producers of antibiotic metabolites effective against *V. cholerae* [98]. Additionally, evidence from large-scale studies suggests that the ratio of bacteriophages to *V. cholerae* during acute infection is inversely correlated with disease severity, indicating that phages may modulate illness outcomes [99]. Bacteriophages are also being investigated as biological alternatives to chemical disinfectants for the treatment of contaminated water sources [45].

**Documented research limitations:** The literature identifies several gaps in the current understanding of alternative approaches, specifically regarding the interactions between phytomedicines and conventional antibiotics [97]. There is limited data on the safety profiles of multi-herb formulations and the ecological stability of biological agents like *Actinomycetes* and bacteriophages in long-term applications [98].

## 4. Discussion

This scoping review highlights that cholera remains a persistent public health challenge in Kenya, driven by the interaction of environmental contamination, inadequate WASH infrastructure, weak surveillance, health-system constraints, and sociocultural factors. Despite advances in epidemiology, molecular surveillance, diagnostics, and intervention design, transmission continues to intensify. Recent WHO reports indicate a worsening cholera burden across Africa, with over 399,000 cases reported between 2022 and mid-2024, and an additional 157,000 cases in early 2025, including Kenya (WHO 2024a, [100]). These trends underscore the urgency of translating research advances into effective, coordinated action.

Across the reviewed literature, cholera-associated mortality in Kenya remains a persistent concern, with case fatality rates frequently exceeding global elimination targets during large outbreaks. Mortality is disproportionately concentrated among children, displaced populations, and residents of informal settlements, particularly during periods of health-system strain, climatic shocks, and delayed outbreak detection. Across studies, mortality is closely linked to delayed care-seeking, limited access to timely rehydration and diagnostics, and weaknesses in surveillance systems that fail to capture deaths occurring outside health facilities. These challenges are especially pronounced in refugee camps, rural areas, and emergency contexts, where under-ascertainment and resource constraints likely contribute to systematic underestimation of cholera-related mortality.

Epidemiological studies consistently document recurrent outbreaks, expanding geographic spread, and disproportionate impact on children, displaced populations, and residents of informal settlements. However, underreporting driven by passive surveillance, inconsistent case definitions, and limited sub-county data continues to obscure disease burden, particularly in refugee settings. While spatial-temporal models have improved hotspot identification and transmission analysis, their routine integration into surveillance and planning remains limited. Future research should incorporate sociocultural factors, antimicrobial resistance, environmental drivers, and vaccine cost-effectiveness into integrated modelling frameworks to support locally tailored responses [70,71,28,3].

Environmental contamination and WASH deficiencies emerge as central structural drivers of cholera transmission. Evidence links outbreaks to unsafe surface water, deteriorating piped systems, flooding, and climate variability. While studies from Lake Victoria and other aquatic systems suggest environmental persistence of *V. cholerae*, recent genomic analyses indicate that regional human mobility may play a larger role in epidemic spread than long-term local reservoirs alone [42,12]. Resolving the relative contributions of environmental persistence versus repeated re-introduction remains a priority, alongside long-term environmental surveillance, improved point-of-use water treatment, and climate-integrated early warning systems [46,54]; Anyamba et al. 2019; [47].

Health-system capacity continues to constrain outbreak detection and response. Although frameworks such as IDSR and the NMCEP have strengthened coordination, shortages in laboratory capacity, trained personnel, and essential supplies persist, particularly in rural and refugee settings. Mathematical modelling studies suggest that integrating WASH indicators with expanded agent-based simulations could improve epidemic prediction and resource allocation [37,38]. Evaluating the cost-effectiveness of oral cholera vaccination in high-risk settings is also critical for sustaining funding and optimising interventions [28,3].

Diagnostic and laboratory limitations remain a major bottleneck. While rapid diagnostic tests and molecular confirmation methods perform well in Kenyan contexts, their uptake is uneven, and antimicrobial susceptibility testing is rarely standardised. Rising antimicrobial resistance further complicates case management. Priorities include expanded AMR surveillance, adoption of cost-effective molecular diagnostics, and evaluation of emerging tools such as wastewater-based epidemiology to strengthen early warning systems [60,58,67].

Whole-genome sequencing has transformed understanding of cholera transmission in Kenya, revealing dominance of the seventh pandemic El Tor lineage alongside locally evolved strains and highlighting the role of mobile genetic elements in antimicrobial resistance [9,77,65]. However, real-time genomic surveillance remains constrained by cost and infrastructure. Future efforts should link genomic, climatic, and epidemiological data and expand research on immune responses and vaccine uptake to better inform control strategies [101,102,71].

Intervention studies show that WASH improvements, vaccination, biological controls, and health education can reduce cholera risk, but impacts are often context-dependent and difficult to sustain. Evidence suggests that aligning WASH interventions with vaccination campaigns improves uptake and effectiveness [84,87,86]. Biological approaches such as bacteriophage therapy show promise but remain largely experimental [45,81].

Social, behavioural, and economic factors strongly shape cholera dynamics. Persistent gaps between knowledge and practice, urban-rural disparities, stigma, and economic constraints delay prevention and care-seeking. Cost-of-illness studies highlight the disproportionate burden on poor households, reinforcing the importance of prevention-focused investments [93]. Culturally grounded health education and long-term evaluation of school-based WASH programmes remain critical [92,88].

Alternative and natural control strategies, including medicinal plants and bacteriophages, demonstrate antimicrobial activity in laboratory studies but lack clinical validation. Future research should identify active compounds, assess safety and interactions with antibiotics, and evaluate feasibility in real-world settings [95,98,97]. Overall, this review shows that while Kenya has generated extensive cholera research across multiple domains, sustained control will depend on integrating scientific advances with robust implementation, multisectoral coordination, and community-driven approaches.

This scoping review has several limitations. First, grey literature and programme reports were excluded, which may have resulted in the omission of operational insights from government and non-governmental implementation programmes. Second, no formal quality appraisal was conducted, consistent with scoping review methodology, which limits the ability to assess the strength or certainty of evidence across individual studies. Third, although the search strategy used the terms "cholera AND Kenya," which was designed to maximise sensitivity for cholera-focused research, it is possible that a small number of laboratory-based studies referring exclusively to *Vibrio cholerae* without explicit mention of cholera were not retrieved.

In addition, while all included studies incorporated data from Kenya, fine-scale geographic synthesis at the county level was not undertaken. Many studies were national in scope, multi-site, or regional, and geographic reporting varied substantially across studies, including the use of counties, former provinces, districts, refugee settlements, or environmental sites. This heterogeneity limited comparability and increased the risk of misclassification if county-level mapping were attempted. As a result, this review focused on thematic, methodological, and temporal mapping of the literature rather than detailed spatial distribution. Future reviews integrating harmonised sub-county data and standardised geographic reporting would be well-positioned to address this gap [103,104,105,106].

## 5. Conclusion and future directions

Cholera control in Kenya will depend on the effective integration of epidemiological surveillance, environmental monitoring, molecular tools, robust health systems, and community-driven interventions. While scientific advances have expanded understanding of transmission dynamics and intervention options, closing implementation gaps remains the central challenge. Strengthening integrated surveillance systems, investing in WASH infrastructure, expanding laboratory and genomic capacity, improving vaccine access, and addressing sociocultural and economic barriers are essential for sustainable progress. Aligning research, policy, and practice within a multisectoral framework offers the best prospect for reducing cholera burden and advancing toward elimination goals.

## Supporting information

**S1 PRISMA Checklist. Preferred Reporting Items for Systematic reviews and Meta-Analyses extension for Scoping Reviews (PRISMA-ScR) checklist, indicating where each recommended reporting item is addressed in this scoping review of cholera research in Kenya.** The PRISMA-ScR checklist is reproduced and used under the Creative Commons Attribution License (CC BY 4.0).
(PDF)

**S1 Table. Summary of included studies on cholera in Kenya: This table summarises the 106 studies in this scoping review on cholera in Kenya.** Each study is assigned a unique "Study ID" and listed by "Title", which includes the authors, year, and journal. The "Summarised Abstract" provides a brief overview of each study's key findings. The "Research Gap" column highlights areas where evidence is limited or missing, while the "Objectives" outline what each study aimed to explore. "Challenges" refer to contextual or practical difficulties encountered, and "Limitations" reflect methodological constraints noted by the authors. The "Future Research" column lists recommended next steps or follow-up studies. "Site" indicates where in Kenya (or beyond) the research was conducted, and "Theme" categorises the study's primary focus, such as Epidemiology, WASH, Surveillance, or Diagnostics.
(XLSX)

## Author contributions

**Conceptualization:** Kevin Wamae, Charles Agoti, George Githinji.

**Data curation:** Kevin Wamae, John Magudha, Agnetor Kakungu.

**Formal analysis:** Kevin Wamae, John Magudha, Agnetor Kakungu, Jolynne Mokaya.

**Funding acquisition:** Samson Kinyanjui, George Githinji.

**Investigation:** Kevin Wamae, John Magudha, Agnetor Kakungu, Steve Aricha, Daniel Langat, Samson Kinyanjui, Jolynne Mokaya, Nicholas R. Thomson, George Githinji.

**Methodology:** Kevin Wamae, John Magudha, Agnetor Kakungu, Steve Aricha, Daniel Langat, Samson Kinyanjui, Jolynne Mokaya, Nicholas R. Thomson, Charles Agoti, George Githinji.

**Project administration:** Kevin Wamae, Samson Kinyanjui, George Githinji.

**Resources:** Kevin Wamae, Samson Kinyanjui, George Githinji.

**Software:** Kevin Wamae, John Magudha, Agnetor Kakungu, George Githinji.

**Supervision:** Kevin Wamae, Samson Kinyanjui, Nicholas R. Thomson, Charles Agoti, George Githinji.

**Validation:** Jolynne Mokaya, Nicholas R. Thomson, Charles Agoti, George Githinji.

**Visualization:** Kevin Wamae, John Magudha, Agnetor Kakungu, Jolynne Mokaya.

**Writing – original draft:** Kevin Wamae, John Magudha, Agnetor Kakungu, Steve Aricha, Daniel Langat, Samson Kinyanjui, Charles Agoti, George Githinji.

**Writing – review & editing:** Kevin Wamae, Jolynne Mokaya, Nicholas R. Thomson, Charles Agoti, George Githinji.

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
