## [Decision Letter · Decision Letter 0]

6 Nov 2025

PGPH-D-25-02924

Cholera in Kenya: a scoping review of current research, evidence gaps and future directions

Dear Dr. Wamae,

Thank you for submitting your manuscript to PLOS Global Public Health. After careful consideration, we feel that it has merit but does not fully meet PLOS Global Public Health’s publication criteria as it currently stands. Therefore, we invite you to submit a revised version of the manuscript that addresses the points raised during the review process.

We look forward to receiving your revised manuscript.

Kind regards,

Tinsae Alemayehu, MD

Guest Editor

Journal Requirements:

1. Please clarify all sources of funding (financial or material support) for your study. List the grants (with grant number) or organizations (with url) that supported your study, including funding received from your institution.

2. State the initials, alongside each funding source, of each author to receive each grant.

3. State what role the funders took in the study. If the funders had no role in your study, please state: “The funders had no role in study design, data collection and analysis, decision to publish, or preparation of the manuscript.”

4. If any authors received a salary from any of your funders, please state which authors and which funders.

2. Please provide separate figure files in .tif or .eps format.

3. supp-1.xlsx is currently uploaded as an 'Other' file type, which is not viewable by reviewers. Please ensure that all files meant for review are uploaded as 'Supporting Information' and include a legend in the manuscript.

Reviewers' comments:

Reviewer's Responses to Questions

**Comments to the Author**

1. Does this manuscript meet PLOS Global Public Health’s publication criteria ? Is the manuscript technically sound, and do the data support the conclusions? The manuscript must describe methodologically and ethically rigorous research with conclusions that are appropriately drawn based on the data presented.

Reviewer #1: Partly

Reviewer #2: Yes

2. Has the statistical analysis been performed appropriately and rigorously?

Reviewer #1: No

Reviewer #2: No

3. Have the authors made all data underlying the findings in their manuscript fully available (please refer to the Data Availability Statement at the start of the manuscript PDF file)?

Reviewer #1: No

Reviewer #2: No

4. Is the manuscript presented in an intelligible fashion and written in standard English?

Reviewer #1: No

Reviewer #2: Yes

Reviewer #1: The review is important to improve outcomes on cholera surveillance and response. However, there are a number of critical issues that must be addressed to ensure the manuscript conforms to the standard of scientific writing and scoping review.

1. Certain sections were ommitted e.g Quality assessment and Data analysis

2. The roles of the authors in the scooping exercise also omitted

3. The results and discussion sections are mixed up. The authors began discussing the findings in the result.

Reviewer #2: Given the ongoing cholera pandemic and its recurrent outbreaks in sub-Saharan Africa, it is commendable that the authors undertook a comprehensive mapping of cholera research in Kenya.

1.For the search strategy, the query “cholera AND Kenya” across all databases is overly restrictive and likely excluded studies using alternative terminology such as “Vibrio cholerae”, “waterborne disease”, or “WASH-related cholera”. I would recommend providing the full keywords, filters and timelines used for each database, to help in reproducibility, as stated in the PRISMA-ScR Checklist (Item 8).

2.Please provide the last search date or timeframe.

3.The authors mentioned the systematic search of five databases, including Google Scholar, Web of Science, PubMed, Embase, and Scopus. However, in the PRISMA flow diagram (Figure 1), there is no data for Google Scholar.

4.The use of Rayyan is recognized. However, reviewer roles, conflict resolution, and data extraction validation are not stated.

5.The authors mentioned the inclusion of non-primary studies, such as reviews, but stated “ineligible study design” as a reason for exclusion in Figure 1. A clarification on this is could be beneficial.

6.For each included study, the authors should present the characteristics of the data charted with respective citations in a table.

7.In section 3.2, the authors provide an informative table which shows the geographic focus of the studies across multiple countries, including Kenya. For a scoping review centered on Kenya, a similar table or map that shows the distribution of studies/ data on the county-level could be added.

8.Themes such as mortality and risk factors of cholera could be explored and discussed further to strengthen the manuscript.

9.The Results-Discussion boundary seems blurred. Discussion begins to appear within “Future directions” paragraphs under each theme. I would recommend that the authors consolidate all “Future directions” into a single Discussion summarising what is known and unknown.

**Do you want your identity to be public for this peer review?** For information about this choice, including consent withdrawal, please see our Privacy Policy .

Reviewer #1: **Yes:** KYENG MERCYKYENG MERCY

Reviewer #2: No

---

## [Decision Letter · Decision Letter 1]

3 Feb 2026

Cholera in Kenya: a scoping review of current research, evidence gaps and future directions

PGPH-D-25-02924R1

Dear %TITLE% Wamae,

We are pleased to inform you that your manuscript 'Cholera in Kenya: a scoping review of current research, evidence gaps and future directions' has been provisionally accepted for publication in PLOS Global Public Health.

Best regards,

Tinsae Alemayehu, MD

Guest Editor

Reviewer Comments (if any, and for reference):

Reviewer's Responses to Questions

**Comments to the Author**

Reviewer #1: All comments have been addressed

publication criteria ? Is the manuscript technically sound, and do the data support the conclusions? The manuscript must describe methodologically and ethically rigorous research with conclusions that are appropriately drawn based on the data presented.

Reviewer #1: Yes

3. Has the statistical analysis been performed appropriately and rigorously?

Reviewer #1: Yes

4. Have the authors made all data underlying the findings in their manuscript fully available (please refer to the Data Availability Statement at the start of the manuscript PDF file)?

Reviewer #1: Yes

5. Is the manuscript presented in an intelligible fashion and written in standard English?

Reviewer #1: Yes

Reviewer #1: All comments have been addressed

**Do you want your identity to be public for this peer review?** For information about this choice, including consent withdrawal, please see our Privacy Policy .

Reviewer #1: **Yes:** KYENG MERCYKYENG MERCY
